# COMPACT BILINEAR POOLING VIA GENERAL BILINEAR PROJECTION

## ABSTRACT

Deep metric learning aims at learning a deep neural network by letting similar samples have small distances while dissimilar samples have large distances. To achieve this goal, the current DML algorithms mainly focus on pulling similar samples in each class as closely as possible. However, the action of pulling similar samples only considers the local distribution of the data samples. It ignores the global distribution of the data set, i.e., the center positions of different classes. The global distribution helps the distance metric learning. For example, expanding the distance between centers can increase the discriminant ability of the extracted features. However, how to increase the distance between centers is a challenging task. In this paper, we design a genius function named the skewed mean function, which only considers the most considerable distances of a set of samples. So maximizing the value of the skewed mean function can make the most significant distance larger. We also prove that current energy functions used for uniformity regularization on centers are special cases of our skewed mean function. At last, we conduct extensive experiments to illustrate the superiority of our methods.

## 1 INTRODUCTION

Deep metric learning (DML) is a branch of supervised feature extraction algorithms that constrain the learned features, such that similar samples have a small distance and dissimilar samples have a large distance. Because having the ability to learn a deep neural network for unseen classes, distance metric learning, i.e., the classes of testing classes do not appear in the training data set, DML are widely used in the applications of image classification & clustering, face re-identification, or general supervised and unsupervised contrastive representation learning Chuang et al. (2020). The goal of DML is to optimize deep neural networks to span its projection space on a surface of a hyper-sphere, in which the semantically similar samples have small distances, and the semantically dissimilar samples have large distances.

This purpose can be formulated as a set of triplets . However, because of the exponential amount of those triplets, distance metric learning needs an additional procedure called as information sample selection, such as hard sample mining and semi-hard sample mining. With the advance of the mining techniques, state-of-the-art metric learning algorithms Wang et al. (2019b); Kim et al. (2020); Roth et al. (2022); Wang & Liu (2021); Schroff et al. (2015); Sun et al. (2020); Deng et al. (2019); Wang et al. (2018) use the log-exp function $q_\lambda(\theta) = log(\sum_{j=1}^{n} e^{\lambda a_i(\theta)})$ Oh Song et al. (2016) to combine the distance metric learning and the information sample selection together. an

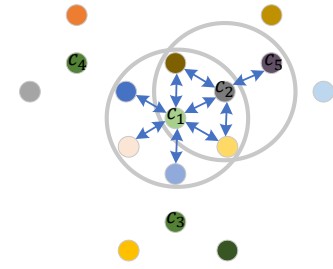

Figure 1: The illustration of assigning the location of centers. $\mathbf{c}_1$ is only pushed away by the six nearest centers. Because the pushing directions are contrary, the position of $\mathbf{c}_1$ is easy to stick. Therefore, the location assignment fails.

Where $\theta$ is the parameter of the deep metric, $\delta_1$ and $\delta_2$ are two tuned parameters, and $\mathcal{S}_i$ and $\mathcal{D}_i$ are the sets of similar and dissimilar samples of the query $\mathbf{x}_i$, respectively.

Although achieving excellent performance, those $log - exp$ function-related algorithms fail to assign different classes' centers. Assigning the locations of class centers facilitates distinguishing the

features from different. For example, if we can let the centers have a significant distance in the features samples, thus the distance of two samples from those two classes will also be increased, making them more easily distinguished.

There are several methods designed to enlarge the distance between centers. The most representative ones are potential energy functions. However, those functions only consider the nearest centers of the query samples. Because the nearest centers are distributed around the query, the pushing actions shown from those nearest centers will let the query sample be stuck, i.e, the position of the query is hard to move in the training stage. As seen from the figure 1, we consider the $\mathbf{c}_1$ as the query centers, and the six centers in the circle are the nearest centers of the center $\mathbf{c}_1$. Because the current energy function only lets the nearest samples push the query samples away, the six nearest samples will let the query sample stuck because the directions of those pushing actions contradict each other. This makes the potential energy function fails to assign the locations of centers.

In this paper, we propose a set of functions named as skewed mean function, which can consider the largest values of a set of samples. In this way, we can let the sample pairs with largest distance be away from each other. Because those samples are on the boundary of the cluster of each sample. Those samples are less to be stuck. Using this finding, we design a regularization term to assign the centers of different classes. The contents include the following aspects:

1. We give a unified framework of proxies-based distance metric learning. From our framework, we summarize the traditional distance metric learning algorithms and the classification-based distance metric learning algorithm together. Therefore, the mathematical proof of their connections gives us a theoretical base for performing ablation experiments to support our comment that all problems of distance metric learning are about the Lipschitz constant.

2. We reveal that the potential energy-based methods have less power to push centers away from each other since they only consider local data information. To alleviate this problem, we adopt the log-exp mean functions and power mean function to design the term to pull the centers of each class. Because we prove the potential energy methods are a special cause of ours, our algorithms have the power to push centers of different classes.

3. We conduct extensive experiments on challenging data sets such as CUB-200-2011, Cars196, Aircraft, and Inshop to illustrate the effectiveness of our algorithms.

**Notation.** $\mathcal{X}^o = \{(\mathbf{x}_i, y_i)\}_{i=1}^{N_1}$ is $C$-class dataset where $\mathbf{x}_i \in \mathbb{R}^{d_1}$ is the $i$-th sample and $y_i \in \{1, \cdots, C\}$ is the label of $\mathbf{x}_i$. $\mathbf{z}_i = f_\theta(\mathbf{x}_i) : \mathbb{R}^{d_1} \to \mathbb{R}^{d_2}$ is a deep neural networks parameterized by $\theta$. The similarity between $\mathbf{x}_i$ and $\mathbf{x}_j$ is denoted as $\mathbf{A}_{ij} = cos(f_\theta(\mathbf{x}_i), f_\theta(\mathbf{x}_j))$. The set of proxies is denoted by $\mathcal{X}^p = \{(\mathbf{w}_k, y_k)\}_{k=1}^{N_2}$ where $\mathbf{w}_k \in \mathbb{R}^{d_2}$ and $y_k \in \{1, \cdots, C\}$ is the corresponding label of $\mathbf{w}_k$. The similarity between $\mathbf{x}_i$ and $\mathbf{w}_j$ is denoted by $\mathbf{B}_{ij} = cos(f_\theta(\mathbf{x}_i), \mathbf{w}_j)$. Because proxy-based DML does not calculate the similarity between samples within $\mathcal{X}$ or $\mathcal{X}^p$, the similar relationship between samples $\mathcal{X}^o + \mathcal{X}^p$ can be depicted by a bipartite graph. For $\mathbf{x}_i \in \mathcal{X}^o$, its similar samples are only in $\mathcal{X}^p$ and denoted as $\mathcal{S}_i^1$. For $\mathbf{w}_i \in \mathcal{X}^p$, its similar samples are only in $\mathcal{X}^o$ and denoted by $\mathcal{S}_i^2$. Likewise, dissimilar sample sets of $\mathbf{x}_i \in \mathcal{X}^o$ and $\mathbf{w}_i \in \mathcal{X}^p$ are denoted by $\mathcal{D}_i^1$ and $\mathcal{D}_i^2$, respectively.

## 2 DISTANCE METRIC LEARNING REVISITED

## 3 SHORTCOMINGS OF DISTANCE METRIC LEARNING

In this section, we adopt the proxy anchor loss as the baseline to analyze current distance metric learning model. The objective function of proxy anchor loss is presented as follows.

$$\mathbb{J} = \frac{1}{|P^+|} \sum_{p \in P^+} log \left(1 + \sum_{\mathbf{x} \in \mathbf{X}_p^+} e^{-\alpha(s(\mathbf{x},\mathbf{p}))-\delta}\right) + \frac{1}{|P|} \sum_{p \in P} log \left(1 + \sum_{x \in X_p^+} e^{\alpha s(\mathbf{x},\mathbf{p})+\delta}\right) \quad (1)$$

where $\delta > 0$ is a margin, $\alpha > 0$ is a scaling factor, $P$ indicates the set of all proxies, and $P^+$ denotes the set of positive proxies of data in the batch. Also, for each proxy $\mathbf{p}$, a batch of embedding vectors $\mathbf{X}$ is divided into two sets: $\mathbf{X}^+$, the set of positive embedding vectors of $\mathbf{p}$, and $\mathbf{X}_p^- = X^p - X_p^+$.

The gradient of the loss function with respect to $s(\mathbf{x}, \mathbf{p})$ is given by

$$\frac{\partial \ell(X)}{\partial s(s,p)} = \begin{cases} \frac{1}{P^+} \frac{-\alpha e^{-\alpha(s(x,p)-\delta)}}{1+\sum_{x' \in X_p^+} e^{-\alpha(s(x,p)-\delta)}}, & \forall x \in X_p^+ \\ \frac{1}{P^+} \frac{-\alpha e^{-\alpha(s(x,p)-\delta)}}{1+\sum_{x' \in X_p^+} e^{-\alpha(s(x,p)-\delta)}}, & \forall x \in X_p^+ \end{cases} \tag{2}$$

In practice, the best performance of distance metric learning algorithms set $\alpha$ a large value. Normally, $\alpha > 32$. According to Eq.(2), we know large $\alpha$ only focuses the farthest similar sample and nearest dissimilar samples in the optimization procedure. Only consider the nearest dissimilar samples means the distance metric leaning only consider the local distribution of the trainging data, and does consider the global information of the training set. As a consequence, the moving of the closest dissimilar sample will be stuck by other ignored dissimilar samples. Because those nearest samples will give each element a pushing force from its anchor or on is its.Therefore, we can claim that the goal of distance metric learning mainly depends on the shrinking of similar samples in each class. In this way, the distance metric learning does not have the power to assign the centers of each classes. Geometrically, pushing centers away from each other will benefit the distinguish samples between different classes. For example, suppose the radius of the cluster region of each class be fixed as $r$, and the gap between two classes be $\delta$. If we let the centers of each class be pushing away from each other, the gap between two classes will also be enlarged, i.e., $\delta + \epsilon$ where $\epsilon$ is the amount increased by the pushing action for centers. In this way, the features extracted by neural networks will be easy to distinguish. Besides, if the centers are not assigned by the algorithm. When we want the gap between two classes still to be $\delta + +\epsilon$, we should shrink samples in the cluster of each class significantly. However, the training samples of distance metric learning are not very much. For example, the widely used dataset in metric learning is CUB-200-2011 has 200 classes with each class 69 samples on average. Compared with dimension of features extracted by neural network, normally being 512 or 1024, the number 69 is very smaller. In this way, it is hard to shrink so less samples in the high dimensional feature space without the overfitting. When the overfitting happens, the performance of distance metric learning will be hurt. Therefore, how to assign the centers of each class is an very important issue.

### 3.1 Shortcomings of Energy Function

Several works are proposed to assign the centers of different classes for classification problem. The well-known ones are the energy function based ones whose formulations are presented as follows.

$$E_{s,d}(\mathbf{w}_i|_{i=1}^C) = \sum_{i=1}^C \sum_{j=1,i \neq i}^C f_s(S(\mathbf{w}_i, \mathbf{w}_j)) = \begin{cases} \sum_{i \neq j} S(\mathbf{w}_i, \mathbf{w}_j)^{-s} & , s > 0 \\ -\sum_{i \neq j} log(S(\mathbf{w}_i, \mathbf{w}_j)) & , s = 0 \end{cases} \tag{3}$$

where $S(\mathbf{w}_i, \mathbf{w}_j)$ is a similarity function between $\mathbf{w}_i$ and $\mathbf{w}_j$. Commonly, there $S(\mathbf{w}_i, \mathbf{w}_j) = |\mathbf{w}_i - \mathbf{w}_j|_2^2$.

Let us calculate the gradient descent of energy function with respective to $s(\mathbf{w}_i, \mathbf{w}_j)$, there is

$$\frac{\partial E_{s,d}(\mathbf{w}_i|_{i=1}^C)}{\partial s(\mathbf{w}_i, \mathbf{w}_j)} = \begin{cases} \sum_{i \neq j}(-s)S(\mathbf{w}_i, \mathbf{w}_j)^{-s-1} & , s > 0 \\ -\sum_{i \neq j} S(\mathbf{w}_i, \mathbf{w}_j)^{-1} & , s = 0 \end{cases} \tag{4}$$

As seen from the above Eq.(4), we know that if $S(\mathbf{w}_i, \mathbf{w}_j)$ is small, the value of gradient descent is large. It means the algorithm would give a large weight to the sample pairs with smaller distance. Thus, the energy function only consider the closest samples of each query, and ignore the farther samples. This have two shortcomings: If only closest samples of each query are considered, so the pushing action on this query sample is easy to be eliminated by the samples around it. Considering there are hundreds of classes in each distance metric learning task, thus, such phenomenons easy encounter. And the centres can not be assigned to the whole surface of the hyper-sphere in the feature space.

In the following content, we design a new mechanism to solve this problem. That is we let the farthest samples of each query to push the query. Because the farthest samples are always located

in the boundary of the region of the features, so when we let the distance between them and query samples, it is hard to be stuck.

### 3.2 DISTANCE METRIC LEARNING SURVEY BY SKEWED MEAN FUNCTIONS

**Definition 1.** *Given a set of numbers $\mathcal{S} = \{s_1, s_2, \cdots, s_N\}$, without loss generality, by setting $0 < s_1 < s_2 < \cdots < s_N$, we can define a $K$ skewed mean of the numbers $\mathcal{S}$ as follows.*

$$M^{[K]}(\mathcal{S}) = \begin{cases} \frac{1}{|K|} \sum_{i=1}^{|K|} s_i & , K < 0 \\ \frac{1}{K} \sum_{i=1}^{K} s_{N-i+1} & , K > 0 \end{cases} \tag{5}$$

*where $K \in \{\pm 1, \pm 2, \cdots, \pm N\}$.*

Obviously, $M^{[-1]} = s_1$, $M^{[1]} = s_N$, and $M^{[N]}(\mathcal{S}) = M^{[-N]}(\mathcal{S}) = \frac{1}{N} \sum_{i=1}^{N} s_i$.

As seen from the definition of the skewed mean functions, it is easy to find the largest value from a set of numbers. If those numbers are the distances between a pair of centers, we can enlarge the skewed mean functions to assign the position of centers. However, those skewed mean function involves the operation of ranking the numbers, which make the skewed mean function is not a continuous function with respective to the distance $S(\mathbf{w}_i, \mathbf{w}_j)$. To solve this problem, we design a series of continuous skewed mean function by introducing the following **Theorem.**

**Theorem 1.** *Given a monotonously continuous increasing function $y = f_\lambda(x) : \mathbb{R}^1 :\rightarrow \mathbb{R}^1$ where $\lambda \in \mathbb{R}^1$, and its inverse function $x = f_\lambda^{-1}(y) : \mathbb{R}^1 :\rightarrow \mathbb{R}^1$, we define a function presented as follows.*

$$b_{\mathcal{S}}(\lambda) = f_\lambda^{-1}(\frac{1}{N} \sum_{i=1}^{N} f_\lambda(s_i)) \tag{6}$$

*We can calculate **the $K$ skewed mean of the numbers** $\mathcal{S} = \{s_1, s_2, \cdots, s_N\}$ defined in Eq.(5) by using Eq.(6) with an appropriate selected $\lambda$, if $b_{\mathcal{S}}(\lambda)$ satisfies the following rules:*

(1) $b_{\mathcal{S}}(\lambda)$ is a monotonously increasing function with respective to $\lambda$

(2) $\lim_{\lambda \rightarrow +\infty} b_{\mathcal{S}}(\lambda) = \max\{s_i\}_{i=1}^{N}$ and $\lim_{\lambda \rightarrow -\infty} b_{\mathcal{S}}(\lambda) = \min\{s_i\}_{i=1}^{N}$

For the Theorem 1, we can give two examples of $f_\lambda(x)$, i.e., $f_\lambda(x) = e^{\lambda x}$ and $f_\lambda(x) = x^\lambda$, which respectively corresponds to

$$b_{\mathcal{S}}(\lambda, a) = \frac{1}{\lambda} \log_a \left( \frac{1}{N} \sum_{i=1}^{N} a^{\lambda s_i} \right)$$
$$b_{\mathcal{S}}(\lambda) = (\frac{1}{N} \sum_{i=1}^{N} (s_i)^\lambda)^{1/\lambda} \tag{7}$$

**Property 1.** *The functions $b(\lambda)$ and $g(\lambda)$ has the following properties:*

(1) *Both $b(\lambda)$ and $g(\lambda)$ are two monotonically increasing functions with respective to $\lambda$;*

(2) $\lim_{\lambda \rightarrow +\infty} b(\lambda) = a_n$ *and* $\lim_{\lambda \rightarrow +\infty} g(\lambda) = a_n$;

(3) $\lim_{\lambda \rightarrow -\infty} b(\lambda) = a_1$ *and* $\lim_{\lambda \rightarrow -\infty} g(\lambda) = a_1$, *thus, there is an appropriate number $\lambda^*$ to let $b(\lambda^*) = a_k$ or $g(\lambda^*) = a_k$ where $a_k$ is the $k$-th largest number in $\{a_i\}_{i=1}^{T}$.*

(4) *Let $a_i = (\boldsymbol{x}_i - \boldsymbol{x})^T \boldsymbol{M} (\boldsymbol{x}_i - \boldsymbol{x})$ where $\boldsymbol{M} \succeq 0 \in \mathbb{R}^{d \times d}$ is a distance metric, $b(\lambda)$ and $-b(-\lambda)$ are convex with respective to the matrix $\boldsymbol{M}$ when $\lambda > 0$.*

**Remark 1.** By using the $K$ skewed mean function, we can automatically select the $k$ largest values of a set of numbers or the smallest values. If we want to assign the centers of different classes in the features space, we should select the center pairs whose distance are large, and let those sample pairs with large distance be pushed away from each other. Because all of those samples are on the surface of a sphere, the distances between those selected sample pairs have a maximal values. In this way, the algorithms will convergence.

### 3.3 REGULARIZATION PUNISHING LARGE SIMILARITY BETWEEN CLASS CENTERS

In this section, we design a term to push centers of classes away from each other. Suppose $\{\mathbf{p}_i\}_{i=1}^{C}$ are centers of classes and the similarity between $\mathbf{p}_i$ and $\mathbf{p}_j$ is denoted as $s(\mathbf{p}_i, \mathbf{p}_j)$. Then, we collect all similarities related to $\mathbf{p}_c$ as a set denoted by $\mathcal{M}_c = \{s(\mathbf{p}_c, \mathbf{p}_i) | j \neq c\}$. In $\mathcal{M}_c$, the $r$-th largest element of is denoted by $v_{\mathcal{M}_c}^{(r)}$. In this way, there is a constraint $v_{\mathcal{M}_c}^{(1)} < \delta_3$ to fulfill the above goal, whose continuous version is presented as follows.

$$R_1 = \frac{1}{\gamma_3} log \left( \sum_{j=1, j \neq i}^{C} e^{\gamma_3(s(\mathbf{p}_i, \mathbf{p}_j) - \delta_3)} \right)$$

$$R_2 = (\frac{1}{C(C-1)} \sum_{i=1}^{C} \sum_{i \neq j} s(\mathbf{p}_i, \mathbf{p}_j)^\lambda)^{\frac{1}{\lambda}} \tag{8}$$

Thus, if we add the above regularization for the metric learning algorithm, we can achieve a new optimization problem which have the ability to constrain the location of centers of different classes.

### 3.4 THE RELATIONSHIP BETWEEN EXISTING METHODS.

Let us introduce a relaxation of the constraint $v_{\mathcal{M}_c}^{(1)} < \delta_3$. Let us combine all $\{\mathcal{M}_c\}_{c=1}^{C}$ to one large set $\mathcal{M} = \bigcup_{c=1}^{C} \mathcal{M}_c$, the set $\{v_{\mathcal{M}_c}^{(1)} | c = 1, \cdots, C\}$ is a subset of $\mathcal{M}$. Therefore, the constraint $v_{\mathcal{M}_c}^{(1)} < \delta_3$ can be relaxed as $\frac{1}{C} \sum_{c=1}^{C} v_{\mathcal{M}_c}^{(1)} < \delta_3$. In this way, by constructing a continuous version of it, we can have an new regularization. Excepting the log-exp mean function, there is another skewed mean function can be used, i.e., $g(\lambda) = (\frac{1}{n} \sum_{i=1}^{n} a_i^\lambda)^{\frac{1}{\lambda}}$. Thus, the new continuous constraint is

$$(\frac{1}{C(C-1)} \sum_{i=1}^{C} \sum_{i \neq j} s(\mathbf{p}_i, \mathbf{p}_j)^\lambda)^{\frac{1}{\lambda}} < \delta_3 \tag{9}$$

If we set the similarity function as the negative distance, the constraint in Eq.(10) is presented as

$$(\frac{1}{C(C-1)} \sum_{i=1}^{C} \sum_{i \neq j} d(\mathbf{p}_i, \mathbf{p}_j)^{-\lambda})^{\frac{1}{-\lambda}} > \delta_3 \tag{10}$$

If we set $\lambda = -1$, the left term in Eq.(10) is the energy based regularization proposed by in Uniformface Duan et al. (2019). If we perform the operation $(x)^{-\lambda}$ on left term of Eq.(10), the minimum hyperspherical energy $\frac{1}{C(C-1)} \sum_{i=1}^{C} \sum_{i \neq j} d(\mathbf{p}_i, \mathbf{p}_j)^{-\lambda}$ is obtained Liu et al. (2018). Because $(x)^{-\lambda}$ is a monotonous decrease function with respect to $\lambda$, the minimum hyperspherical energy has the same goal of Eq.(10).

For the regularization term used in Uniformface, to let $\lambda = -1$ will reduce the flexibility of the algorithm to suit different types of data, because we know the $\lambda$ is a parameter related to the class number $C$. Different from Uniformface, the minimum hyperspherical energy term has a parameter $\lambda$ on $s(\mathbf{p}_i, \mathbf{p}_j)$. However, $s^{-\lambda}(\mathbf{p}_i, \mathbf{p}_j)$ will be very large with a relative small $\lambda$ if $s(\mathbf{p}_i, \mathbf{p}_j)$ is small. Such a large value will make its coefficient in the objective function hard to tune. Thus, in practice, $\lambda$ could not be selected too large. Actually, $\lambda$ is set to $0, 1, 2$ in Liu et al. (2018). This means minimum hyperspherical energy term also lacks enough flexibility to deal with different types of samples.

Besides the flexibility, the above mentioned two methods should calculate $C(C-1)/2$ times similarity, which is extremely large when the class number of the task is large. So many calculation will make the gradient update very slow. For example, in the face recognition, the class number can be more than $690K$, so such the terms used in Uniformface and minimum hyperspherical energy term will cost plentiful computational resources. However, in our algorithm, we consider the $\{\mathbf{p}_i\}_{i=1}^{C}$ as nodes in the bipartite graph. Similar to samples in $\mathcal{X}$, we can also only select small part of $\{\mathbf{p}_i\}_{i=1}^{C}$ to construct the objective function. In this way, our algorithm can save a lot of computational resource.

## 4 EXPERIMENTAL RESULTS

In this section, our method is evaluated and compared to current state-of-the-art methods on the four benchmark datasets for deep metric learning. We also investigate the effect of hyperparameters and embedding dimensionality of our loss to demonstrate its robustness.

### 4.1 DATASETS

We employ CUB-200-2011Wah et al. (2011), Cars-196Krause et al. (2013), Stanford Online Product (SOP)Oh Song et al. (2016) and In-shop Clothes Retrieval (In-Shop) datasets Liu et al. (2016) for evaluation. For CUB-200-2011, we use 5864 images of its first 100 classes for training and 5,924 image of the other classes for testing. For Cars-196, 8054 images of its first 98 classes are used for training and 8131 images of the other classes are kept for testing. For SOP, we follow the standard dataset split in Oh Song et al. (2016) using 59551 images of 11,318 classes for training and 60,502 images of the rest classes for testing. Also for In-Shop, we follow the setting in Oh Song et al. (2016) using 25882 images of the first 3,997 classes for training and 28,760 images of the other classes for testing; the test set is further partitioned into a query set with 14,218 images of 3,985 classes and a gallery set with 12,612 images of 3,985 classes.

### 4.2 IMPLEMENTATION DETAILS

**Embedding network:** For a fair comparison to previous work, the inception network Ioffe & Szegedy (2015) with batch normalization pre-trained for ImageNet classification is adopted as our embedding network. We change the size of its last fully connected layer according to the dimensionality of embedding vectors, and $L_2$-normalize the final output.

**Training:** In every experiment, we employ AdamW optimizer Loshchilov & Hutter (2017), which has the same update step of Adam Kingma & Ba (2014) yet decays the weight separately. Our model is trained for 40 epochs with initial learning rate $10^{-4}$ on the CUB-200-2011 and Cars-196, and for 60 epochs with initial learning rate $6 \cdot 10^{-4}$ on the SOP and In-shop. The learning rate for proxies is scaled up 100 times for faster convergence. Input batches are randomly sampled during training.

**Image setting:** Input images are augmented by random cropping and horizontal flipping during training while they are center-cropped in testing. The default size of cropped images is $224 \times 224$ as in most of previous work, but for comparison to HORDE Jacob et al. (2019), we also implement models trained and tested with $256 \times 256$ cropped images.

### 4.3 ABLATION EXPERIMENT ON DIFFERENT METHODS

To demonstrate the importance of the neighborhood parameter learning, we conduct an ablation study on CUB-200-2011. Since the outer objective is a standard metric learning based on the log-exp function, therefore, we could instead it with the objective function of other type of metric learning algorithm, such as multi-similar loss Wang et al. (2019b), N-pair lossSohn (2016), lifted-structure lossOh Song et al. (2016), Proxy-nca loss Movshovitz-Attias et al. (2017) and the adaptive neighborhood metric learning Song et al. (2021).

The reason why we select those three methods to conduct the ablation experiment, is they are the special cases of the adaptive neighborhood metric learning Song et al. (2021). We adopt the reformulated metric learning methods as the outer objective function of our methods, our bi-level learning framework could solve the neighborhood parameters and the metric parameter according to the algorithm **??**. We utilize the dataset CUB-200-2011 to training those methods, and adopt the Recall@1 to evaluate the performance of them. The results are shown in the Table 1. As seen from the Table 1, with the help of the neighborhood parameter learning, those well-known metric learning algorithms could be improved further in terms of performance.

Table 1: Performance on the CUB-200-2011 of the three state-of-the-art methods and their improved versions with 512 dimension.

| $Recall@k$ | | 1 | 2 | 4 | 8 |
|---|---|---|---|---|---|
| Lifted structure loss$^{512}$ | Original | 45.4 | 58.4 | 69.5 | 79.5 |
| | bi-level | 47.1 | 60.3 | 71.6 | 81.8 |
| Multi-similar loss$^{512}$ | Original | 49.2 | 61.9 | 67.9 | 72.4 |
| | bi-level | 52.7 | 65.4 | 68.3 | 75.7 |
| N-pairs loss$^{512}$ | Original | 43.6 | 56.6 | 68.6 | 79.6 |
| | bi-level | 46.8 | 60.7 | 72.6 | 83.8 |
| Proxy-NCA loss$^{512}$ | Original | 44.6 | 53.6 | 68.6 | 84.6 |
| | bi-level | 47.8 | 62.7 | 72.5 | 87.8 |
| DANML$^{512}$ | Original | 65.4 | 76.8 | 85.7 | 90.7 |
| | Improved | 67.6 | 79.1 | 88.2 | 93.4 |

## 4.4 COMPARISON TO STATE-OF-THE-ART METHODS

With further comparison the performance of our method with state-of-the-art techniques on image retrieval task, we conduct the proposed methods on the CUB-200-2011, Car-196, Stanford Online Product (SOP) and In-shop Clothes Retrieval (In-Shop) datasets. We adopt the recall@k as the metric to evaluate the performance of the related metric learning methods. The result are shown in Table 2-4.

As shown in Table 2, our DANML improves Recall@1 by $1.9\%$ on the CUB-200-2011, and $1.5\%$ on the Cars-196 over the recent state-of-the-art multi-similarity loss. This may be because the logistic loss function is more powerful than the linear function for generalization. Meanwhile, for recently proposed method Circle Loss, our DANML outperforms it about $0.9\%$ on the CUB-200-2011 and $2.2\%$ on the Cars-196 dataset. Compared with ABE which is an ensemble method with a much heavier model, our method achieves a higher Recall@1 by $7.0\%$ improvement on the CUB-200-2011 and $0.4\%$ on the Cars-196 dataset.

For the Stanford Online Products (SOP) and the In-Shop Clothes Retrieval (In-Shop), as seen from Tables 4 and 3, our method outperforms multi-similarity loss by $1.7\%$ on the In-Shop dataset and by $0.4\%$ on the SOP dataset, respectively. Furthermore, when compared with ABE, our method increases Recall@1 by $3.6\%$ and $2.8\%$ on the In-Shop and SOP dataset, respectively. For the Circle Loss which is a recent state-of-the-art method on SOP dataset, our DANML achieves a better performance about $1.6\%$ on it.

## 4.5 IMPACT OF HYPERPARAMETERS

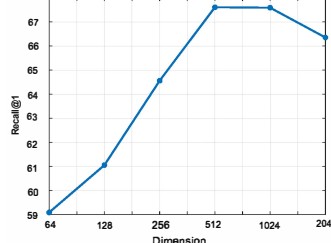
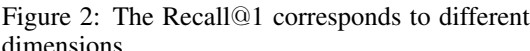

Figure 2: The Recall@1 corresponds to different dimensions.

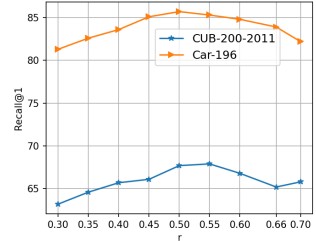

Figure 3: The Recall@1 corresponds to different $r$.

**Embedding dimension:** The dimension of embedding vectors is a crucial factor that controls the trade-off between speed and accuracy in image retrieval systems. We thus investigate the effect of embedding dimensions on the retrieval accuracy in our Bi-level metric learning framework. We

Table 2: Recall@K(%) performance on CUB-200-2011 dataset and Cars-196 dataset. Superscript denotes embedding size.

| $Recall@K(\%)$ | CUB-200-2011 | | | | | | Cars-196 | | | | | |
|---|---|---|---|---|---|---|---|---|---|---|---|---|
| | 1 | 2 | 4 | 8 | 16 | 32 | 1 | 2 | 4 | 8 | 16 | 32 |
| Clustering[64]Oh Song et al. (2017) | 48.2 | 61.4 | 71.8 | 81.9 | - | - | 58.1 | 70.6 | 80.3 | 87.8 | - | - |
| ProxyNCA[64]Movshovitz-Attias et al. (2017) | 49.2 | 61.9 | 67.9 | 72.4 | - | - | 73.2 | 82.4 | 86.4 | 87.8 | - | - |
| Smart Mining[64]Harwood et al. (2017) | 49.8 | 62.3 | 74.1 | 83.3 | - | - | 64.7 | 76.2 | 84.2 | 90.2 | - | - |
| Margin[128]Wu et al. (2017) | 63.6 | 74.4 | 83.1 | 90.0 | 94.2 | - | 79.6 | 86.5 | 91.9 | 95.1 | 97.3 | - |
| HDC[384]Wu et al. (2017) | 53.6 | 65.7 | 77.0 | 85.6 | 91.5 | 95.5 | 73.7 | 83.2 | 89.5 | 93.8 | 96.7 | 98.4 |
| HTL[512]Ge (2018) | 57.1 | 68.8 | 78.7 | 86.5 | 92.5 | 95.5 | 81.4 | 88.0 | 92.7 | 95.7 | 97.4 | 99.0 |
| ABIER[512]Opitz et al. (2018) | 57.5 | 68.7 | 78.3 | 86.2 | 91.9 | 95.5 | 82.0 | 89.0 | 93.2 | 96.1 | 97.8 | 98.7 |
| ABE[512]Kim et al. (2018) | 60.6 | 71.5 | 79.8 | 87.4 | - | - | 85.2 | 90.5 | 94.0 | 96.1 | - | - |
| Multi-similarity loss[512]Wang et al. (2019b) | 65.7 | 77.0 | 86.3 | **91.2** | 95.0 | 97.3 | 84.1 | 90.4 | 94.0 | 96.5 | 98.0 | 98.9 |
| Hardness-aware[512]Zheng et al. (2019) | 53.7 | 65.7 | 76.7 | 85.7 | - | - | 79.1 | 87.1 | 92.1 | 95.6 | - | - |
| Circle Loss[512]Yifan Sun (2020)66.7 | 77.4 | 86.2 | **91.2** | - | - | 83.4 | 89.8 | **94.1** | 96.5 | - | - |
| Ranked list loss[512]Wang et al. (2019a) | 61.3 | 72.7 | 82.7 | 89.4 | - | - | 82.1 | 89.3 | 93.7 | 97.7 | - | - |
| Ours[512] | **67.6** | **79.1** | **86.4** | **91.2** | **97.1** | **98.1** | **85.6** | **92.1** | **94.1** | **97.7** | **98.1** | **99.3** |

Table 3: Recall@K(%) performance on SOP dataset. Superscript denotes embedding size.

| $Recall@k$ | SOP | | | |
|---|---|---|---|---|
| | 1 | 10 | 100 | 1000 |
| Clustering[64]Oh Song et al. (2017) | 67.0 | 83.7 | 93.2 | - |
| ProxyNCA[64]Movshovitz-Attias et al. (2017) | 73.7 | - | - | - |
| Smart Mining[64]Harwood et al. (2017) | 49.8 | 62.3 | 74.1 | - |
| Margin[38]Wu et al. (2017) | 72.7 | 86.2 | 93.8 | 98.0 |
| HDC[384]Wu et al. (2017) | 69.5 | 84.4 | 92.8 | 97.7 |
| HTL[512]Ge (2018) | 74.8 | 88.3 | 94.8 | 98.4 |
| ABIER[512]Opitz et al. (2018) | 74.2 | 86.9 | 94.0 | 97.8 |
| ABE[512]Kim et al. (2018) | 76.3 | 88.4 | 94.8 | 98.2 |
| Multi-similarity loss[512]Wang et al. (2019b) | 78.2 | 90.5 | 96.0 | 98.7 |
| Hardness-aware[512]Zheng et al. (2019) | 68.4 | 83.5 | 92.3 | - |
| Ranked list loss[512]Wang et al. (2019a) | 79.8 | 91.3 | 96.3 | - |
| Circle Loss[512]Yifan Sun (2020) | 78.3 | 90.5 | 96.1 | 98.6 |
| Ours[512] | **79.9** | **92.1** | **96.4** | **98.9** |

Table 4: Recall@K(%) performance on In-Shop dataset. Superscript denotes embedding size.

| $Recall@k$ | In-Shop | | | | | |
|---|---|---|---|---|---|---|
| | 1 | 10 | 20 | 30 | 40 | 50 |
| FashionNet[4096]Oh Song et al. (2017) | 53.0 | 73.0 | 76.0 | 77.0 | 79.0 | 80.0 |
| HDC[384]Wu et al. (2017) | 62.1 | 84.9 | 89.0 | 91.2 | 92.3 | 93.1 |
| HTL[512]Ge (2018) | 80.9 | 94.3 | 95.8 | 97.2 | 97.4 | 97.8 |
| ABIER[512]Opitz et al. (2018) | 83.1 | 95.1 | 96.9 | 97.5 | 97.8 | 98.0 |
| ABE[512]Kim et al. (2018) | 87.3 | 96.7 | 97.9 | 98.2 | 98.5 | 98.7 |
| Multi-similarity loss[512]Wang et al. (2019b) | 89.7 | 97.9 | 98.5 | 98.8 | 99.1 | 99.2 |
| Ours[512] | **90.1** | **98.2** | **98.9** | **99.0** | **99.3** | **99.4** |

test our loss with embedding dimensions varying from $64$ to $1,024$ following the the experiment in Wang et al. (2019b), and further examine that with 32 embedding dimension. The result of analysis is quantified in Figure 2, in which the retrieval performance of our loss is compared with that of MS loss Wang et al. (2019b). The performance of our loss is fairly stable when the dimension is equal to or larger than $128$. Moreover, our loss outperforms MS loss in all embedding dimensions, and more importantly, its accuracy does not degrade even with the very high dimensional embedding unlike MS loss.

**Parameter $r$ in our method:** We also investigate the effect of the hyperparameter $r$ of our method on the Cars-196 dataset and CUB-200-211. The results of our analysis are summarized in Figure 3, in which we examine Recall@1 of the proposed bi-level metric learning loss by varying the values of the parameter $r \in \{0.3, 0.35, 0.4, 0.45, 0.5, 0.55, 0.6, 0.66, 0.7\}$. For CUB-200-211 and Cars-196, the results suggest that when $r$ near $0.55$ and $0.45$, the proposed bi-level metric learning achieve the best performances, respectively. The results indicate the performance of the proposed bi-level methods is sensitive to the parameter $\lambda$, so we should carefully chose $r$ in the proposed bi-level distance metric learning algorithm. The $r$ is the gap between two negative classes which determines the lower-bound of the Lipschitz constant of the learned deep neural network network. That is why the performance of the proposed methods is sensitive to the value of $r$.

## 5 CONCLUSION

In this paper, we reveal that learning the position of centers for each class is very important to metric learning. However, current potential energy-based regularization has less ability to constrain the position of centers because it considers the nearest centers of each query center. The pushing actions given by the nearest centers on the query center contradict each other. To overcome this shortcoming, we design a function named skewed mean function, which can be used to calculate the most considerable distances of a set of numbers. Using the skewed mean function, we give new center regularization, which considers center pairs with farthest centers. The conducted experiments illustrate the effectiveness of our proposed method.

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
