# OpenReview forum: "Improve distance metric learning by learning positions of class centers"
_ICLR.cc/2023/Conference — Submitted to ICLR 2023_

### Official Review · Reviewer_CgdK · 2022-10-20

**Confidence:** 5
**Correctness:** 1
**Technical Novelty And Significance:** 2
**Empirical Novelty And Significance:** 2
**Recommendation:** 3

**Clarity, Quality, Novelty And Reproducibility:**

The quality of the paper is poor and way behind the ICLR conference standards. There are to many annotation mistakes and the paper is not written well. The novelty is also limited in my opinion. The experiments can be reproduced if the authors share the codes.

**Strength And Weaknesses:**

The only strength of the paper is the reported better accuracies compared to the existing methods. It seems the authors proposed a new method, but the novelty is limited.
However, there are many weaknesses regarding the paper and the proposed methodology. These can be summarized as follows:
i) First of all, the paper is written very badly. The mistakes start at the first page, the authors forget to include an equation, but they describe the parameters of the missing equation. Also, there are many annotation mistakes as well. The following statements describing the motivations do not make any sense at all: “Because those samples are on the boundary of the cluster of each sample. Those samples are less to be stuck. Using this finding, we design a regularization term to assign the centers of different classes.” “… to support our comment that all problems of distance metric learning are about the Lipschitz constant.”
ii) The claim that the most existing methods only consider the local relations between classes and ignore the global ones is incorrect. For example, the UniformFace discussed in the paper targets to learn equi-distributed representations for face recognition. It considers all the relations between class centers. However, the distance metric is updated based on the closest class centers, and it makes perfect sense the closest samples are the hardest ones for separation. The classes far from each other can be already separated easily.
iii) There are methods that update the centers based on the feature representations such as the methods using center loss [1] and discriminative centers [2]. They also consider both local and global relations among the classes and centers are updated dynamically. These methods should be also discussed in the paper.
iv) The authors criticize the methods such UniformFace and methods that target diverse deep neural network weights that are uniformly distributed on a hypersphere. Yet, they do not make any experimental comparison. Especially, comparing the proposed method to UniformFace on large-scale face recognition problems is necessary to support their claims.
v) There are also mistakes in the tables. There are better accuracies outperforming the authors’ proposed method that must be indicated in bold characters in the tables.

References
[1] Y. Wen, K. Zhang, Z. Li, and Y. Qiao. A comprehensive study on center loss for deep face recognition. International Journal of Computer Vision, 127:668–683, 2019.
[2] B. Uzun, H. Cevikalp and H. Saribas, "Deep Discriminative Feature Models (DDFMs) for Set Based Face Recognition and Distance Metric Learning," in IEEE Transactions on Pattern Analysis and Machine Intelligence, 2022.

**Summary Of The Paper:**

In this paper, the authors claim that existing distance metric learning methods update the distance metric based on local relationships and ignore the more global ones (however, this claim is not right). The authors argue that updating distance metric based on the closest class centers has some drawbacks and they claim that the furthest distances must be also used for distance metric learning. To maximize the distances between the class centers they utilize skewed mean function, but it is not clear how this information is used. The authors also show some relations between their proposed methodology and the existing ones and they claim that the existing methods focusing on uniform distributions through energy functions are special cases of their proposed method. Lastly, the authors conduct some experiments on different datasets and report better accuracies compared to the existing methods in the literature.

**Summary Of The Review:**

This is badly written paper. There are frequent English Grammar mistakes, typos and annotation mistakes. Some of the equations is not complete and there are inconsistencies among the notations. All these things made reading the paper extremely difficult. Also, some of the claims are not correct. This paper is clearly not ready and it needs a serious proofread and rewriting. Experimental evaluations is also not complete. Therefore, the paper must be rejected in my opinion.

---

### Official Review · Reviewer_SAjL · 2022-10-25

**Confidence:** 4
**Correctness:** 3
**Technical Novelty And Significance:** 2
**Empirical Novelty And Significance:** 2
**Recommendation:** 3

**Clarity, Quality, Novelty And Reproducibility:**

The quality and clarity of this work is poor, and the originality of the work is fair.

**Strength And Weaknesses:**

Strength:
+ This paper designs the skewed mean function used in deep metric learning to only considers the most considerable distances of a set of samples.

Weaknesses:
- The title of this paper is "COMPACT BILINEAR POOLING VIA GENERAL BILINEAR PROJECTION", which is not appropriate for the content.
- The Section "2 DISTANCE METRIC LEARNING REVISITED" is not complete.
- The results of the proposed method on Table 2 & Table 3 are not SOTAs.
- The paper is not well-written.

**Summary Of The Paper:**

This paper considers the global distribution of the dataset in deep metric learning, and introduces the skewed mean function to only considers the most considerable distances of a set of samples. The paper also proves that current energy functions are special cases of the skewed mean function. Extensive experiments are conducted to illustrate the superiority of the proposed method.

**Summary Of The Review:**

See major weaknesses in Strength And Weaknesses.

---

### Official Review · Reviewer_tGz6 · 2022-10-25

**Confidence:** 5
**Clarity, Quality, Novelty And Reproducibility:** Terrible.
**Correctness:** 2
**Technical Novelty And Significance:** 2
**Empirical Novelty And Significance:** 2
**Recommendation:** 1

**Strength And Weaknesses:**

Strength: the studied problem is interesting.

Weaknesses: the contribution is not clear, and the article is an unfinished work. In section 2, nothing is revisited but the title. The writing is terrible. There are too many spelling and grammatical errors in Section 3. In general, this paper is far from acceptance.


**Summary Of The Paper:**

This paper presents a method to expand the distance between centers of different classes. However, the contribution is not clear, and the article is an unfinished work. In section 2, nothing is revisited but the title. The writing is terrible. There are too many spelling and grammatical errors in Section 3. In general, this paper is far from acceptance.

**Summary Of The Review:**

As I stated above.

---

### Decision · Program_Chairs · 2023-01-20

**Decision:**

Reject

**Justification For Why Not Higher Score:**

Paper is not clear, and the work is incomplete

**Justification For Why Not Lower Score:**

The paper is recommended for rejection

**Metareview: Summary, Strengths And Weaknesses:**

The paper presents a skewed mean function to increase distance between clusters inorder to improve classifications.

The work is incomplete and the paper is not so clear. There are several technical issues as indicated by Reviewer CgdK.

**Summary Of Ac-Reviewer Meeting:**

NIL